# Alfalfa Spring Black Stem and Leaf Spot Disease Caused by *Phoma medicaginis*: Epidemic Occurrence and Impacts

**DOI:** 10.3390/microorganisms12071279

**Published:** 2024-06-24

**Authors:** Yanru Lan, Wennan Zhou, Tingyu Duan, Yanzhong Li, Cory Matthew, Zhibiao Nan

**Affiliations:** 1State Key Laboratory of Herbage Improvement and Grassland Agro-Ecosystems, Lanzhou University, Lanzhou 730020, China; lanyr19@lzu.edu.cn (Y.L.);; 2Key Laboratory of Grassland Livestock Industry Innovation, Ministry of Agriculture and Rural Affairs, Lanzhou University, Lanzhou 730020, China; 3Engineering Research Center of Grassland Industry, Ministry of Education, Gansu Tech Innovation Centre of Western China Grassland Industry, Lanzhou University, Lanzhou 730020, China; 4College of Pastoral Agriculture Science and Technology, Lanzhou University, Lanzhou 730020, China

**Keywords:** *Phoma medicaginis*, alfalfa, symptoms, pathogenicity, control

## Abstract

Alfalfa spring black stem and leaf spot disease (ASBS) is a cosmopolitan soil-borne and seed-borne disease caused by *Phoma medicaginis*, which adversely affects the yield, and nutritive value and can stimulate production of phyto-oestrogenic compounds at levels that may adversely affect ovulation rates in animals. This review summarizes the host range, damage, and symptoms of this disease, and general features of the infection cycle, epidemic occurrence, and disease management. ASBS has been reported from over 40 countries, and often causes severe yield loss. Under greenhouse conditions, reported yield loss was 31–82% for roots, 32–80% for leaves, 21% for stems and 26–28% for seedlings. In field conditions, the forage yield loss is up to 56%, indicating that a single-cut yield of 5302 kg/ha would be reduced to 2347 kg/ha. *P. medicaginis* can infect up to 50 species of plants, including the genera *Medicago*, *Trifolium*, *Melilotus*, and *Vicia*. ASBS is more severe during warm spring conditions before the first harvest than in hot summer and cooler winter conditions, and can infect alfalfa roots, stems, leaves, flowers, pods, and seeds, with leaf spot and/or black stem being the most typical symptoms. The primary infection is caused by the overwintering spores and mycelia in the soil, and on seeds and the cortex of dead and dry stems. The use of resistant cultivars is the most economical and effective strategy for the control of ASBS. Although biological control has been studied in the glasshouse and is promising, chemical control is the main control method in agriculture.

## 1. Introduction

Alfalfa (*Medicago sativa* L.) is used worldwide as a major high protein forage worldwide to support animal-based industries. Attributes of alfalfa include its high nutrient content, good forage palatability, forage nutritive value and longevity, and its adaptation to a wide range of cultivation conditions [1,2]. Alfalfa also can be used in the phytoremediation of heavy-metal-contaminated soils [2,3] and the improvement of soil fertility and physico-chemical properties on account of its large root system [4]. Alfalfa is widely introduced and cultivated in more than 80 countries around the world, with the total planted area exceeding 32 million hectares [5,6]. The United States has 8.90 million hectares of alfalfa, with an estimated product value of over USD 9.3 billion in 2018 [7]. Other countries or regions with significant areas of alfalfa are China with 4.71 million hectares in 2017 producing about 32.71 million metric tons (quote from [8]), Argentina with 4.7 million hectares in 2006 [9], Canada with 4 million hectares [10], and Turkey with 0.66 million hectares in 2020 [11].

Diseases are a major constraint to the health, growth, and sustainability of alfalfa crops and cause significant losses of production worldwide. Pathogen-induced direct impacts on alfalfa systems include decreased forage and seed yield [12,13], reduced nutritive value [14], and impacts on animal health and reproduction through the secondary metabolite, coumestrol, in the forage produced [15,16], as well as increased costs and side effects of disease control programs. Alfalfa leaf spot diseases, including *Leptosphaerulina* spp., anthracnose disease, and *Pseudopeziza* spp., occur throughout the entire growing season. Inparticular, alfalfa spring black stem and leaf spot disease (ASBS) caused by *Phoma medicaginis*, is a cosmopolitan soil-borne [17,18] and seed-borne disease [18,19]. The disease symptoms typically include leaf spot, black stem, seedling blight, and crown and root rot. The disease causes the most severe yield reduction in the first spring harvest [20]. The taxonomy of *P. medicaginis* has recently been revised, with a new name, *Ascochyta medicaginicola*, proposed in 2015 [21], but here, we continue to use the former name for ease of comparison with previous studies of the same fungus. This review focuses on the taxonomic classification of the pathogen, its host range, and symptoms and damage caused by ASBS, but also provides generalized information on infection spread, conditions predisposing to epidemics, resistance mechanisms, control measures, and directions for further research.

## 2. The Pathogen

### 2.1. Classification

*Phoma* is considered an anamorphic fungus and belongs to the class Dothideomycetes, order Pleosporales, and family Didymellaceae of the Ascomycota [21]. *Phoma* taxonomy is set out in detail in the publication *Phoma Identification Manual*, which recognizes 223 specific and infra-specific taxa of *Phoma*, classified into nine *Phoma* sections based on morphology and culture characteristics [22]. However, the classification of species in *Phoma* and allied genera is still controversial. *Phoma* and related genera were revised to Didymellaceae, a newly established family, based on sequence data of the 18S rDNA and the 28S rDNA regions in 2009 [23]. The generic delimitation of *Ascochyta*, *Didymella*, *Epicoccum*, and *Phoma* genera in Didymellaceae is still not clear. A further study published in *Studies in Mycology* in 2015 showed by combining multi-locus phylogenetic analyses based on ITS, LSU, rpb2, and tub2, and morphological observations, that *Ascochyta*, *Didymella* and *Phoma* are three different genera [21].

ASBS was first reported in the United States nearly 80 yr ago [24]. Countries where ASBS has been found include Netherlands [25], Canada [26], Australia [17], and others. The pathogen causing ASBS was first named *Ascochyta imperfecta* Peck in 1912 Quoted from [25]. In 1960, a study showed that the pathogens causing ASBS and red clover (*Trifolium pratense* L.) black stem disease are the same, and *A. imperfecta* was considered to be the correct name for ASBS [27]. However, the most correct name for the fungus was considered to be *P. medicaginis* Malbr. & Roum, not *A. imperfecta*, nor *P. herbarum* f. *medicaginum* and a clear description was published in 1965 comparing the fungus causing alfalfa, red clover, and pea (*Pisum* sp.) black stem disease [25]. For the pathogen causing ASBS, two varieties are recognized, based on the morphological and physiological characteristics, *P. medicaginis* var. *medicaginis* and *P. medicaginis* var. *macrospora*, based on the morphological and physiological characteristics in 2002 [28]. At room temperature, the variety *P. medicaginis* var. *macrospora* shows strong resemblance, or only slightly larger conidia dimensions than the variety *P. medicaginis* var. *medicaginis.* At low temperature, the size of conidia ((2.8–)6.3–11.1(–27.8) × (1.4–)2.1–2.9(–5.8)) µm, the number of septate conidia (10–63% with 1–3 septate conidia) and pathogenicity (more aggressive) of *P. medicaginis* var. *macrospora* exceeded those of the variety *P. medicaginis* var. *medicaginis,* for which corresponding conidia sizes were (4.2–)5.7–7.2(–12.7) × (1.4–)2.1–2.3(–3.5) µm [21,28]. The conidial morphological characteristics of the two *P. medicaginis* varieties cultured at 25 °C and 5 °C were observed in the course of research by the authors (Figure 1). However, cluster analysis placed this pathogen in the genus *Ascochyta*. Moreover, the two varieties were located in the same branch without any difference in four sequenced loci in an analysis that combined multi-locus phylogenetic analyses based on *ITS, LSU, rpb2*, and *tub2* in 2015 [21]. Therefore, the two varieties were each given another new name, *A. medicaginicola* var. *medicaginicola* and *A. medicaginicola* var. *macrospora* [21].

### 2.2. Culture Characteristics

A range of media are suitable for *P. medicaginis* culture. *P. medicaginis* colonies on oatmeal agar, potato carrot agar, potato dextrose agar, potato sucrose agar, malt extract agar, and czapek dox cultured at 25 °C are depicted in Figure 2. The optimum growth media are V8 medium, alfalfa extract medium, and glucose peptone medium. Among these, the mycelia were reported to grow fastest on V8 medium, while sporulation was reported to be most prolific on alfalfa extract medium, and the spore germination rate was highest on glucose peptone medium [29]. In addition, pycnidia formation was favored by moderate vitamin fortification of the medium [30]. The formation of pycnidia, the formation of conidia, growth of hyphae, and the germination of conidia were optimized at 30 °C, 20 °C, 20 °C, and 25 °C, respectively on PDA in dark conditions. Reported lethal temperatures for mycelia and conidia were 50 °C (10 min) and 44 °C (10 min), respectively [29,31]. The effect of culture temperature on *P. medicaginis* colony diameter observed in research by the authors is shown in (Figure 3). Germination of conidia and development of pycnidia is rapid and profuse at over 80–100% relative humidity [30]. Constant light can increase the numbers of pycnidia and conidia [29]; however, pycnidia development was the same in cultures exposed to constant light and alternate darkness and light as for those kept in the dark [29,30]. *P. medicaginis* can grow in media with pH ranging from 3–12, while the optimum for sporulation and spore germination is pH 6 [29].

Various isolates of *P. medicaginis* reacted differently to different sugars, to different sugar concentrations, to the balance between sugar and nitrogen, and (in the early stages) to the inclusion of different amino acids in synthetic media [30]. Polysaccharides were superior to the monosaccharides and disaccharides with respect to the numbers of conidia produced. However, cultures with monosaccharides and disaccharides produced more pycnidia than cultures with polysaccharides. Generally, the formation of pycnidia and conidia was favored by nitrate more than by ammonium nitrogen sources. The average number of conidia and pycnidia was greatest when the nitrogen source was NH_4_NO_3_. All amino acids tested appeared to be useful nitrogen sources for the production of pycnidia and none was especially suitable for conidia production [31]. The most suitable carbon sources for mycelia growth, sporulation, and spore germination were sucrose, glucose, and fructose, and suitable nitrogen sources were beef extract, peptone, and yeast extract. The form of nitrogen supplied was not limiting, but the addition of nitrogen to the medium favored pycnidia development more than the addition of sugar [30].

Crystals produced beneath the colony are a typical feature of the culture of *P. medicaginis* in all isolates [32]. Bryoid, dendritic crystals of Brefeldin A, can appear after about 2 or 3 weeks on malt agar in darkness at 20 °C [32]. Brefeldin A inhibited the growth of the common phylloplane fungus mycelium [33].

## 3. Host Ranges of *P. medicaginis*

The pathogen causing ASBS has been reported to have a wide host range [34]. The reported hosts are mainly from the genera *Medicago*, *Trifolium*, and *Melilotus*, with specific reports for those genera including the *Medicago* species *M. sativa* L. ssp. *falcata* (L.) Arcang. [35], the *Trifolium* species *T. pratense* L., *T. hybridum* L., and *T. incarnatum* [27,35], and the *Melilotus* species *M. officinalis* (L.) Lam. and *M. albus* Medik. [35,36]. Other reported hosts include the *Vicia* species *V. sativa* L., *V. villosa* Roth., and *V. faba* L. [35], and *Lathyrus sylvestris* L. [34]. All of these genera belong to the legume family, Fabaceae. Non-leguminous hosts include *Calluna vulgaris* (L.) Hull. [37], and *Chamaedaphne calyculata* (L.) Moench. [38]. Research into the pathogenicity of isolates from alfalfa on other hosts showed that the alfalfa isolates have greater pathogenicity towards the host from which they were isolated than towards other hosts [25,34]. However, one study [39] that tested the pathogenicity of *P. medicaginis* isolates against eight commonly cultivated legume species suggested that *P. medicaginis* has a more restricted host range, limited to *M. sativa*. The host response to the fungus is a complex process, that also has a time dimension and is affected by many factors. These include the environment, the inoculation route, and the host and pathogen genetic strain. Therefore, the host range of *P. medicaginis* remains incompletely defined.

## 4. Symptoms in Infected Alfalfa Tissues

ASBS is a systemic disease, infecting all organs and tissues of alfalfa plants. This is illustrated by a summary of publications reporting details of artificial infection for research purposes in various countries (Table 1). The pathogen can attack leaves [25,33,40,41], petioles [42,43], stems [42,44], crowns [45,46], roots [47], and seeds [35,48] in field conditions. Symptoms of *P. medicaginis* infection on leaves, stems, and roots are shown in Figure 4. Similar results are seen under controlled conditions, and pod and peduncle lesions were also observed in the greenhouse [19]. Leaf spotting and stem blackening are the symptoms most often seen in the field. The symptoms expressed in different organs are inconsonant. For infected leaves, initially, small fawn, brown, purple, or black dark lesions develop. As the spots enlarge and coalesce, the spot may be nearly round, elliptic, or irregular. Sometimes the black spots of the leaf undersides are more obvious than those on the upper surfaces. Leaf symptoms are particularly evident in more humid climates [26,40,42,49]. Under suitable conditions, a large number of small black spots are found in the middle of the disease spot. These are pycnidia of the pathogen [44]. For infected petioles, the most common symptoms are dark brown or black spots, rounded, elliptical, or irregular in shape [43]. On infected pods, dark brown spots are observed, and the pods are typically wrinkled and empty [20]. For infected seeds, no clear symptoms are observed except seed shriveling, but a large number of shallow or dark brown spots (pycnidia) are produced when the seeds are cultured on wet filter paper in Petri dishes. Seedlings from diseased seeds often die within about a week of germination [20]. For infected stems, the initial *P. medicaginis* infection spots typically elongate vertically. Stem lesions, shedding of diseased leaves, blackening of lower stems of plants through coalescence of stem lesions, and even death of small shoots in severe cases, are all symptoms which may be observed [44]. For infected roots, brown discoloration, withering, wilting, and collapse are observed [41]. After prolonged exposure to moist conditions, root neck rot and rot of the upper part of taproot are late symptoms of the disease [25,45,47].

## 5. The Effect of *P. medicaginis* on Productivity of Alfalfa

*P. medicaginis* can cause complete defoliation and premature death of very susceptible medics [53]. It is a main cause of yield loss and losses in forage quality in alfalfa, particularly in lush stands during wet weather or in irrigated fields [53]. In extreme cases, it can destroy the whole plant, resulting in total loss of seed yield [14].

Under greenhouse conditions, plant-size and vigor-related traits such as stem number, crown diameter, stem and root dry weight, and number of axillary buds are typically all reduced, compared to controls, in *P. medicaginis*-infected plants, regardless of whether tissues inoculated were wounded or unwounded. In various trials, weight reductions were, respectively, 31–82%, 32–80%, 21%, and 26–28% for roots, leaves, stems, and seedlings (Figure 1). For example, the plant root and foliage dry weights were reduced by 50% and 47%, respectively, at 30 days after inoculation by root wounding. The same traits were reduced by 31% and 37%, after inoculation of unwounded roots [47]. Even higher yield reductions have been reported. In one experiment, inoculation of wounded roots reduced root dry weights by up to 82% and foliage dry weights by 80% [45]. Other examples include 32% and 21% dry weight reductions in leaves and stems, respectively, compared with healthy plants, after inoculation with *P. medicaginis* [51], and a foliage dry weight reductions in alfalfa seedlings of 28% and 26%, at 30 days after crown wounding or stubble inoculation, respectively [46].

In field conditions, there are reports from Canada, New Zealand, Australia, and the United States, of leaf spot diseases causing forage yield losses of 6%–40% (quote from [54]). The more severe the disease, the greater the yield loss. Another field study revealed that the dry-matter yield of alfalfa was reduced from about 5302 kg/ha to 2347 kg/ha when disease severity rating increased from level 0 to level 3.4 [44]. It is worth noting that the loss is more serious in older stands. Also, plant density and sward persistence of alfalfa decreased gradually as the diseased plants died year by year.

Protein levels are an important trait when assessing forage quality. Leaf spots decrease the forage feed quality by limiting the performance of photosystem II, leading to reduced carbohydrate and protein contents [55]. Leaf crude protein content of 20 alfalfa cultivars was reported to be reduced by 11–60%, compared with healthy leaves, in 19 of the cultivars after inoculation with *P. medicaginis* [54].

In addition, a widely recognized negative feed quality effect of *P. medicaginis* on alfalfa is the stimulation of production of coumestans, phyto-oestrogenic compounds that can reduce the ovulation rate and reproductive performance of animals grazing affected forage, especially sheep [16,56]. A study on *M. polymorpha* var. *brevispina* showed that there was significant positive correlation at the end of the growing season between the disease severity and coumestrol content of dry stems and pods, with coumestrol content increased by up to 75% in stems and 321% in pods of affected plants [15] (Figure 5). The content of coumestrol was up to 1995 mg/kg in stems of *M. murex* cultivar Zodiac infected with *P. medicaginis*, while it was 145 mg/kg in stems of healthy *M. murex* cultivar Zodiac [57]. Coumestrol is affected by intrinsic plant factors, such as pathogen isolate [57], *Medicago* cultivar [57], plant growth stage [58], and the position of the affected plant tissue in the canopy [59].

## 6. Disease Epidemiology

### 6.1. Overwintering

*Phoma medicaginis* depends mainly upon pycnidia and pycniospores for the completion of its life cycle. Most studies of the life cycle of the pathogen are from the 20th century. The mature pycnidia or mycelia develop and overwinter in the cortical tissues below the epidermis in old or dead plant lesions, especially on dead stems remaining over winter [30]. It has been reported that mycelium in alfalfa tissues in the soil remains viable for up to two years [60,61]; pycniospores remain in dry stem material for up to 50 months [30]. Pathogen overwintering in these ways is presumed to be the primary source of inoculum for outbreaks of ASBS in spring (Figure 6).

### 6.2. Infection Process

*Phoma medicaginis* infection usually starts with spore germination, producing one or two germ tubes [62]. The germinating structure penetrates the host tissue epidermis when conditions are suitable. Wound invasion and penetration through stomata of the leaves are the main invasion pathways of *P. medicaginis* [40]. For example, severe infection was more likely to occur following inoculation of wounded plant tissues than unwounded tissues, and inoculation of intact crowns produced no rot symptoms in repeated tests [46], but wounding significantly increased the extent and frequency of root necrosis following root inoculation [45,47]. There is a latent or incubation period after the pathogen first invades tissues. Therefore, *P. medicaginis* can at times be isolated from symptomless tissues [33].

The pathogen colonizes and advances after the successful invasion of host tissues. The hyphae of *P. medicaginis* grow along the grooves between epidermal cells of alfalfa roots, and the cell wall and matrix of epidermal cells close to the *P. medicaginis* hyphae are often degraded by fungal enzyme activity. As the infection progresses, the root epidermal cells begin to deteriorate [62]. In that study, root necrosis was confined to the stele and often extended to the zone of lateral root proliferation, occasionally causing necrosis in lateral roots. Internal necrosis also extended into basal portions of stems through the first internode. When cut ends of stubble were inoculated with *P. medicaginis*, necrosis advanced down the stub from the infection point, with the dead stub later becoming bleached or tan behind the blackened leading edge of the infection. Necrosis stopped at the node, and the stub portion below the node became chlorotic. Pycnidia developed in the bleached area of the stub. Internal necrosis extended from the bleached area of the stub into the crown, and infection also caused the death of axillary buds [46].

Crown inoculation with *P. medicaginis* caused a black necrosis that extended up into stem bases through one internode and down into upper taproots and occasionally into lateral roots. Extensive discoloration of the vascular tissues was observed in the stem, and pathogen colonization of the vascular tissue of the upper taproot was detected [45,46]. Mature pycnidia formed at the original sites of infection, while the pathogen spread within the leaf tissue along the veins until the entire leaf was affected. When pods were inoculated with *P. medicaginis*, the fungus colonized the inner surface of the seed coat and produced some pycnidia, and thus became established on developing seeds [20].

### 6.3. Epidemiology of ASBS

Environmental factors, especially humidity and temperature, are important determinants of spore germination and/or the penetration of hyphae into the plant tissues, and hence ASBS epidemiology. Moisture significantly promotes development, exudation, the release of pycniospores from pycnidia, and their spread [30]. ASBS is more severe during warm spring conditions before the first cut is taken, for example in late May or June in the northern hemisphere (mean daily temperature 12 °C) or autumn conditions than in the hot summer, such as in July and August (mean daily temperature 20 °C) and cooler winter conditions [63,64,65], but it is more frequent in the spring than in the fall, and occurrence appears to be related to the frequency of rain rather than to mean rainfall [49]. One study reported that *P. medicaginis* exhibited more severe leaf disease at day/night temperatures of 21/16 °C, followed by 18/13 °C, and less severe symptoms at 15/10 °C [63]. *P. medicaginis* develops most rapidly at low temperature and in the presence of free moisture on plants from dew or rain [19,64]. High humidity for a longer period of time (7 days) increased severity of symptom occurrence and promoted rapid disease progression [63]. However, another study reported that the incidence of *P. medicaginis* was not significantly related to weather variables, such as monthly mean temperature and total rainfall [66]. It has also been observed that *P. medicaginis* failed to infect alfalfa in a very dry season. Another study found that short periods of exposure to short-wave ultraviolet light resulted in increased infection, while longer exposures (up to 200 s) caused a progressive reduction in infection and spore viability, as assessed by a detached leaf culture technique [67].

## 7. Disease Assessment

Timely and accurate disease assessment is a necessary first step in the control of alfalfa diseases [68]. Remote sensing offers a major advantage in that plant canopies (sampling units) can be objectively and repeatedly analyzed, both nondestructively and noninvasively. The technology is used in plant pathology to assess the amount of disease injury in plant populations [69]. Remote sensing assessment is used in a broad range of alfalfa foliar disease to monitor mixed infections of several alfalfa diseases such as ASBS caused by *P. medicaginis*, summer black stem and leaf spot caused by *Cercospora medicaginis*, common leaf spot caused by *Pseudopeziza medicaginis*, Leptosphaerulina leaf spot caused by *Leptosphaerulina briosiana*, and Stemphylium leaf spot caused by *Stemphylium* spp. [68,70]. The serious alfalfa root rot disease caused by *Phymatotrichopsis omnivora* can also be monitored using remote sensing techniques [71]. Remote sensing is rarely used for the detection of ASBS only, but rather for simultaneous detection of multiple diseases. In addition, in order to assess disease occurrence more accurately, an accurate and rapid tool to detect and differentiate other alfalfa pathogens in a single PCR reaction, such as the *Sclerotinia* species [72], *Phymatotrichopsis omnivora* [73], *Phoma sclerotiodes* [74], and *Paraphoma radicina* [75], is generally used to formulate appropriate control programs before a major outbreak occurs.

## 8. Disease Control

Strategies for control of ASBS in various *Medicago* species include selection of cultivars with increased resistance to *P. medicaginis*, fungicidal spray applications, biological control, and adoption of cultural practices that reduce *P. medicaginis* infection. The most economic and effective disease control method is the screening and breeding of resistant varieties.

### 8.1. Host Resistance—Induced Disease Resistance

A high level of systemic resistance to plant disease, which will not be passed on to the next generation of plants, can in some cases be stimulated by external factors, including various physical, chemical, and biological elicitors [76]. Some such measures have been reported to induce disease resistance in a number of alfalfa–pathogen interactions. Accumulation of pathogenesis-related (PR) proteins is known to be induced in alfalfa by physical elicitors (harvesting, wounding, and heat treatments), chemical elicitors (abscisic acid and ethylene treatments), and biological elicitors (*Xanthomonas campestris* pv. *alfalfae*) [77]. In addition, the expression of defence genes against *P. medicaginis* infection, linked to chitinase activity, phenylalanine metabolism and photosynthesis enhancement was also induced by inoculating with arbuscular mycorrhizal fungi [78]. Leaves smeared with a 1 mg/mL jasmonic acid solution inoculation with *P. medicaginis* exhibited a disease incidence reduction of 37.2% and a disease index reduction of 49.6%, compared to leaves without jasmonic acid treatment [79].

### 8.2. Resistant Cultivars

The cultivars Ramsey (with moderate ASBS resistance) and Ranger (ASBS susceptible) were reported as *M. sativa* standard reference cultivars [80]. In a field study, cultivars Hi-phy and Vernal had, respectively, the highest and the lowest disease severity rating [44]. A number of historical studies have also elucidated resistance of the annual *Medicago* spp. to ASBS [81,82]. Among the three *Medicago* species, *M. truncatula*, *M. ciliaris*, and *M. polymorpha*, *M. ciliaris* is the least susceptible to *P. medicaginis* infection at the species level [83]. In a study by Barbetti, genotypes with very high levels of resistance to ASBS were *M. sphaerocarpos* GRC5659.4.1 and SAD10069, *M. murex* GRC87.1, GRC707, and GRC708, *M. truncatula* Z771, and *M. solerolii* DZA3180.1, all of which had stem disease scores ≤1.0 (scale 0 to 10) at the end of the growing season [84]. The expression of resistance has been associated with the activity of specific genes coding for the production of isoflavonoid phytoalexins [81]. Some ASBS-resistant genes and proteins were identified by carrying out a de novo genome assembly of the ASBS-resistant *M. truncatula* accession SA27063. The identified candidate genes further indicate that application of molecular technology in alfalfa to enhance disease resistance is promising [85]. Overall, resistant varieties escape infection by inhibition of spore germination, penetration, and mycelium development [40], linked to activities of superoxide dismutase, catalase, isoflavone reductase, phenylalanine ammonia-lyase, and chalcone synthase in the hosts [29,86].

The contents of phyto-oestrogenic compounds (coumestrol and 4’-O-methylcoumestrol) have a significant positive relationship with disease severity parameters of the susceptible cultivars, but resistant cultivars were also found to contain high levels of phyto-oestrogenic compounds. Therefore, breeding alfalfa cultivars with resistance to *P. medicaginis* should not only be performed under conditions moderately favorable for development of ASBS to ensure a disease challenge during selection, but also consider the levels of phyto-oestrogenic compounds. Cultivars with less propensity for phytoestrogen production like *M. truncatula* cv. Caliph can provide useful source material when breeding for reduced phytoestrogen production [57,82,87]. The same principle likely applies also for *M. sativa*.

### 8.3. Biological Control

Biological control methods are also available for the control of ASBS. Leaf lesion areas pre-colonized by *P. medicaginis* failed to develop, when those lesions were sprayed with conidial suspensions of *Cladosporium cladosporioides* or *Penicillium citrinum* [88], with AM fungi *Funneliformis mosseae* or *Rhizophagus intraradices*, with the rhizobium *Sinorhizobium medicae* [89,90], and with *Bacillus licheniformis* [91] or with *B. subtilis* [92]. Also, *Streptomyces* inoculated onto the seed at the time of planting or onto leaves also showed potential to reduce leaf spot caused by *P. medicaginis* incidence and index under greenhouse conditions [88].

The AM fungus *R. intraradices* has been reported to reduce disease incidence by 39.48% and the disease index by 56.18%, thus ameliorating the effects of *P. medicaginis* infection on alfalfa, resulting in increased plant levels of total nitrogen and total phosphorus and enhanced alfalfa growth [78,90]. *R. intraradices* can also enhance alfalfa resistance to aphid infestation, providing a basis for strategies to manage pathogens and herbivore pests [93]. In addition, *Sinorhizobium meliloti* strain 10.16/R6 has been reported to be antagonistic to *P. medicaginis* and to decrease the length of black stem necrosis and plant mortality of *M. truncatula* plants infected with *P. medicaginis* by 65% and 80% [94]. Many species of *Bacillus* have been used to control alfalfa disease, and they show broad-spectrum antifungal activity. *B. licheniformis* strains also can control ASBS. In one experiment [89], root browning, stem browning, and plant death decreased, respectively, by 88.8%, 71.7%, and 100%, compared to untreated plants. The protective ability of *B. licheniformis* was attributed to its high chitinolytic activity and antifungal properties [89]. *B. subtilis* strain L194 also alleviates disease symptoms by reducing germination of *P. medicaginis* conidia [95]. *B. subtilis* strain UD1022 is directly antagonistic not only towards *P. medicaginis*, but also towards *Collectotrichum trifolii* and *Phytophthora medicaginis* [92]. *B. subtilis* can also control alfalfa root rot disease. For example, the *B. subtilis* strain CG-6 exerted an 87.33% growth inhibition against *Fusarium oxysporum* and the protective ability of *B. subtilis* was attributed to secreted antibacterial enzymes, siderophores, and indoleacetic acid, and phosphate solubilization [96]. *B. amyloliquefaciens* against alfalfa anthracnose disease (*Colletotrichum truncatum*) can produce bacillomycin D and fengycin [97]. Nonribosomal peptide (NRP) surfactins, the largest class of *Bacillus* spp. antibiotics, were regarded as playing an important role in the antagonism of Bacillus spp. towards *P. medicaginis* [92].

### 8.4. Cultural Practice

Modified cultural practices provide another option for ASBS disease control. Specifically, timely mowing (or grazing) and adequate fertilization assist ASBS control. For example, foliar fertilization with growth stimulators Bionat (a foliar organic fertilizer containing 6.9% N and other nutrients) [98], Atonik (a mixture of three nitrophenolates), and Cropmax (a preparation of fermentation metabolites) applied to field alfalfa plantings reduced ASBS disease incidence by 5%, compared to unfertilized crops [99]. We found no published work directly linking other aspects of cultural practice to the prevention of ASBS. Many reports of agricultural husbandry measures exist for other alfalfa diseases. Sclerotinia crown and stem rot of alfalfa caused by *Sclerotinia sclerotiorum* can be controlled by fall burning. In one study, an intense surface fire reduced numbers of sclerotia in an alfalfa seed field by more than 95% [100]. Other husbandry-oriented control methods, including grazing, fertilizer application, rotations, and seed health, are also strategies available to manage and reduce alfalfa disease [13]. For example, green manure application and crop rotation also significantly reduced alfalfa root rot and yield [101].

### 8.5. Chemical Control

Chemical prevention is the major control measure for ASBS disease. Historically, benomyl [54,102], mancozeb [103], glyphosate applied to glyphosate-tolerant alfalfa [104], chlorothalonil [105], and dithane M–45 [106] have been successfully applied to control ASBS under greenhouse or field conditions. Disease control can be quantified in terms of a decrease in the area under a disease progress curve and is typically linked to an increase in dry matter yield and crude protein level of mature, healthy alfalfa plants. In laboratory trials, benomyl and propiconazole were highly effective at inhibiting *P. medicaginis* growth, with EC50 values ranging from 1.0 to 14.4 ng/mL [102]. In a field trail, ASBS incidence on leaves of *M. sativa* was lower by 48–52% in benomyl and propiconazole treatments in Alberta [54]. Glyphosate reduces symptom severity of ASBS because, although glyphosate is a herbicide, the plant enzyme pathways it inhibits are also found in fungi. Hence, as noted above, the spraying of glyphosate on roundup-resistant alfalfa has been shown to control ASBS [104]. Solamargine caused a 50% growth inhibition of *P. medicaginis* at 60 µM (at pH 7) [107]. We found no other published work directly linking other fungicides to the prevention of ASBS. Many fungicides are supplied for controlling alfalfa foliar diseases including ASBS. It has been reported that annual alfalfa yield was increased by 7.5 to 13.0% annually in Arizona when alfalfa was treated with the fungicide chlorothalonil to control foliar diseases [108,109]. Also, alfalfa yield was increased by 7.4% per cutting when treated with the fungicide mancozeb in Kansas [110]. However, the fungicide cupric hydroxide registered for use on alfalfa was reported to be the least effective of those tested in controlling foliar pathogens of alfalfa [68]. Chemical options provide effective control of alfalfa diseases in some circumstances. However, the costs and concerns about potential side effects of agricultural chemicals preclude their extensive use in many countries. In recent years, concern about pesticide residues, environmental pollution, and food safety has curtailed the use of pesticides. Benomyl use in the USA, propiconazole and mancozeb use in the European Union, and glyphosate use in Costa Rica have been phased out because of emerging toxicity concerns.

## 9. Concluding Perspectives

Increased understanding of biological characteristics and the life cycle of *P. medicaginis* are key to the formulation of appropriate control programs before a major outbreak occurs. Rapid progress in the identification and classification of the pathogen has provided a necessary science capability for research into ASBS. The understanding of the pathogenicity of *P. mediaginis* and its disease symptoms is built up from the combined insights from greenhouse testing, detached-leaf tests, and field observations and experiments. Understanding of the ASBS disease cycle and epidemiology offers further prospects of developing novel approaches to management of the disease. The breeding of resistant varieties and chemical control are the main measures for combatting ASBS. The intensification of human activities, and the abuse of fungicides, and in some areas, increased temperatures related to climate change, appear to have combined to allow the pathogen to develop resistance to fungicides and a greater capacity to attack previously resistant varieties. Climate change and human activities have increasingly become major factors influencing crop health and disease levels. Each link of the disease infection cycle, including spore germination, infection and transmission, and mycelium colonization, is sensitive to the prevailing climatic environment. Local resistant varieties should be bred, taking account of local climatic conditions. A future research priority is clarification of the culture characteristics and intrinsic differences at the gene level between *P. medicaginis* varieties. Also needed is more detailed information on levels of mycotoxins in ASBS-compromised alfalfa forage, and the associated animal health implications when utilizing alfalfa forage.

## Figures and Tables

**Figure 1 microorganisms-12-01279-f001:**
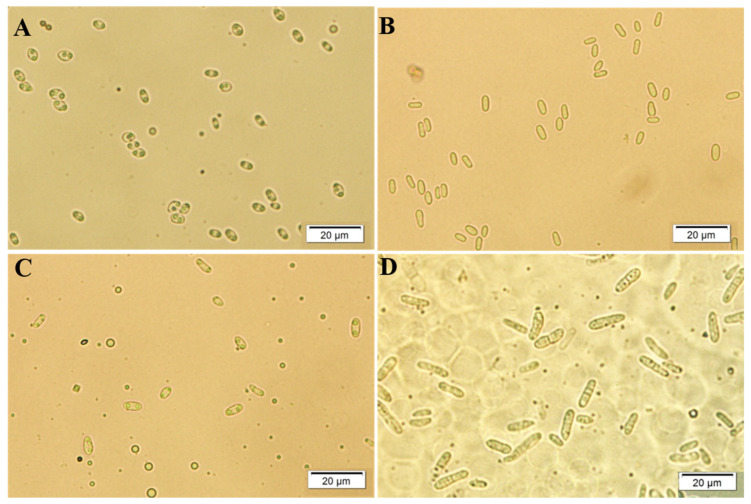
The morphological characteristics of *Phoma medicaginis* var. *medicaginis* (**A** and **C**, at left) and *Phoma medicaginis* var. *macrospora* (**B** and **D**, at right) cultured at 25°C (**A** and **B**, above) and 5 °C (**C** and **D**, below) (from unpublished data by Yanru Lan).

**Figure 2 microorganisms-12-01279-f002:**
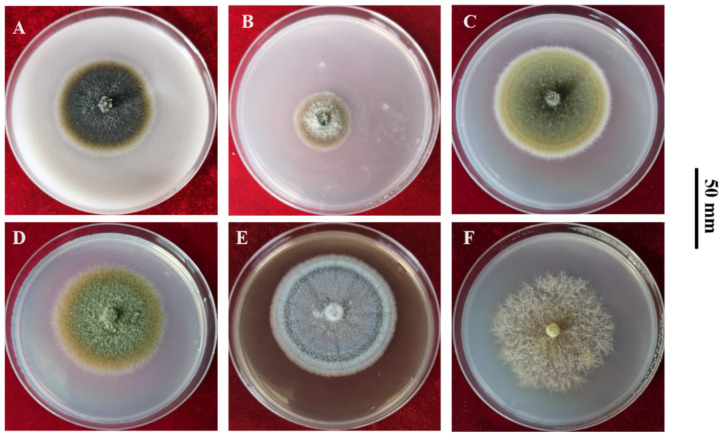
Authors’ unpublished data from an experiment comparing growth of *Phoma medicaginis* at 25 °C on different culture media. Methodology details for the experiment are given in Appendix A. The photos for each culture medium are representative of 4 physical replicates and all depict the same isolate: **A**, cultured on oatmeal agar; **B**, potato carrot agar; **C**, potato dextrose agar; **D**, potato sucrose agar; **E**, malt extract agar; and **F**, czapek dox agar (from unpublished data by Yanru Lan).

**Figure 3 microorganisms-12-01279-f003:**
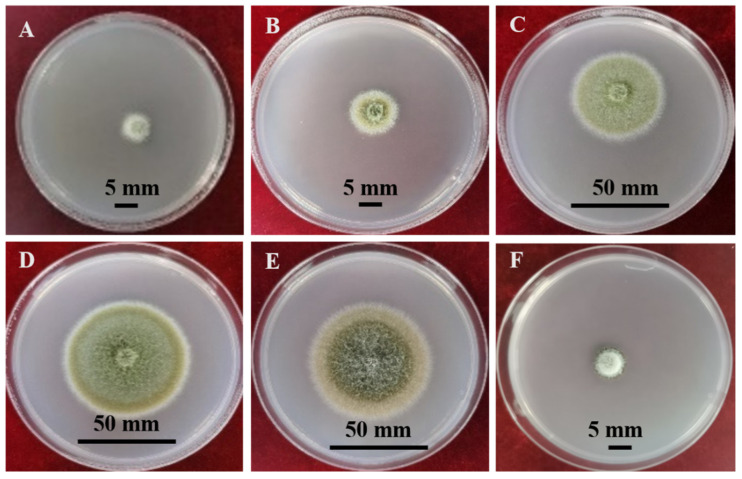
The effect of different temperatures on the *P. medicaginis* colony cultured on PDA for 6 days. **A**, **B**, **C**, **D**, **E** and **F**; *Phoma medicaginis* colony cultured at 5 °C, 10 °C, 15 °C, 20 °C, 25 °C, and 30 °C, respectively (from unpublished data by Yanru Lan).

**Figure 4 microorganisms-12-01279-f004:**
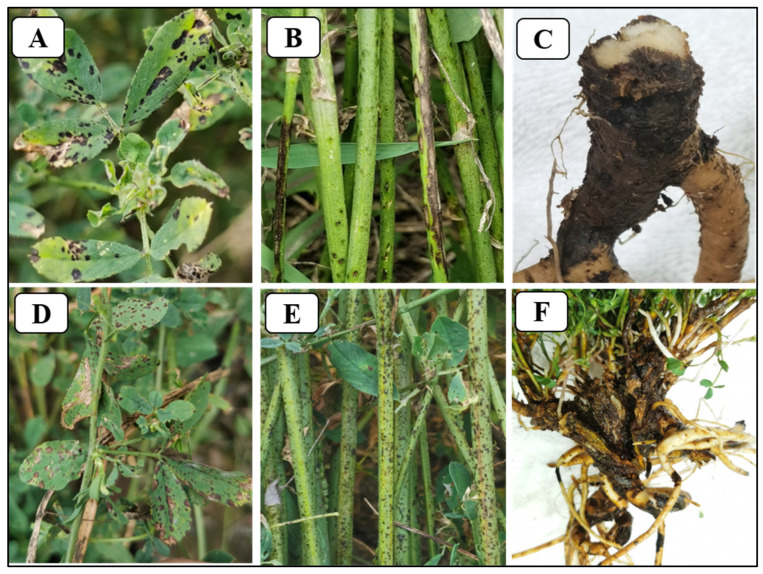
Symptoms associated with *Phoma medicaginis* infection of alfalfa leaves, stems, and roots observed in the fields by the authors. **A** and **D**, small black spots on the leaf; **B** and **E**, stem blackening; **C** and **F**, root necrotic lesions and rot (from unpublished data by Yanru Lan).

**Figure 5 microorganisms-12-01279-f005:**
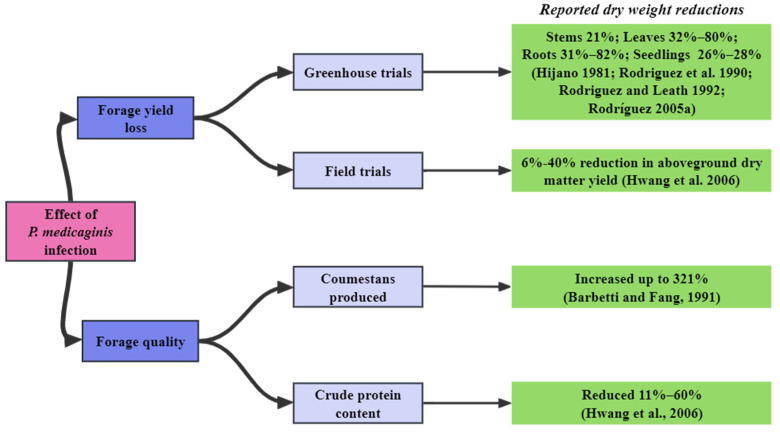
The effect of *P. medicaginis* on alfalfa dry-matter yield and selected measures of forage nutritive value [15,45,46,47,51,54].

**Figure 6 microorganisms-12-01279-f006:**
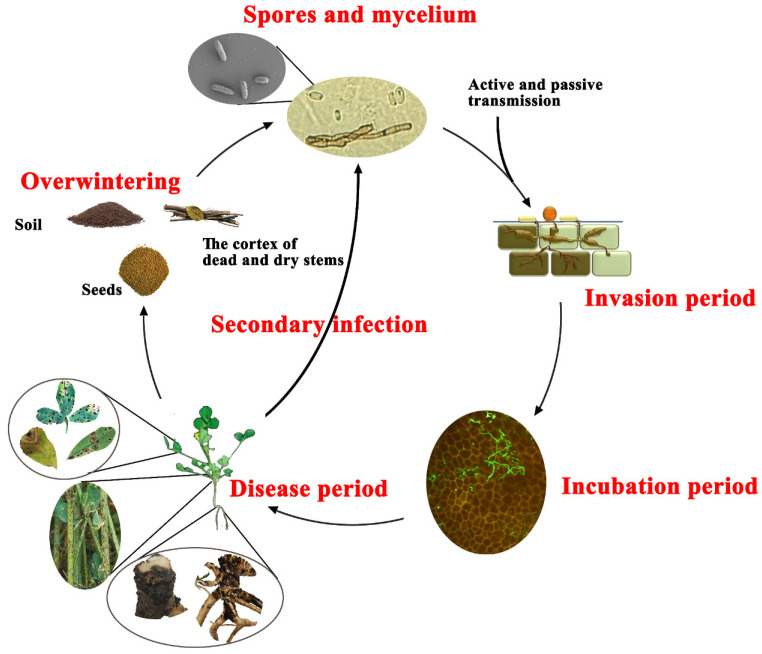
Schematic representation of the presumed disease cycle of *P. medicaginis* on alfalfa. Infection is initiated when germ tubes produced by spores invade wounded tissues (roots, stems, or leaves) or the host stomata. There follows an incubation period during which *P. medicaginis* intercellular hyphal growth occurs. Under suitable conditions, symptoms subsequently emerge, including root and crown necrosis, blackening of stems, leaf spots with black dark necrotic lesions, or unclear or clear concentric rings along the leaf margins and tips. The infected tissues in turn provide a new source of inoculum that can infect healthy plants. Infected seeds, soil and debris (especially the cortex of dead and dry stems) provide refugia where the pathogen overwinters. In the following season, the overwintering pathogen completes the infection cycle.

**Table 1 microorganisms-12-01279-t001:** Summary details of experimental infection of alfalfa with Phoma medicaginis in various countries.

Country	Plant Tissue Isolated From	Inoculated Tisues	Inoculation Method	Symptom(s)	Reference(s)
Canada	Seeds	Excised leaves	Spraying spore suspension	Lesion	[35]
-	-	Pods, peduncles	Spraying homogenized agar cultures of *P. medicaginis*	Lesion	[20]
Netherlands, USA and Canada	Seeds, stems, leaves	Roots and detached leaves	Roots: dipping in a spore suspension; Detached leaves: spraying with spore suspension	Roots: yellow lower leaves and brownish-black stems and roots; leaves: leaf spot	[25]
-	-	Leaves	Spraying spore suspension	Slight chlorosis appeared 3 days after inoculation, and typical black leafspots formed by the 6th day.	[50]
USA	-	Leaves and stems	Stems: spraying spore suspension at 2.5 × 10^7^ spores/mL after rubbing with silicon carbide; Leaves: spraying spore suspension at 2.5 × 10^5^ spores/mL	Foliar and stem lesions	[51]
USA	-	Leaves and stems	Spraying spore suspension at 2 × 10^6^ spores/mL	Foliar and stem lesions	[44]
-	*M. truncatula*	Leaves, stems and petioles	Spraying spore suspension at 7.0 × 10^6^ spores/mL	Leaf lesions; petioles collapsed	[42]
USA	Seeds	Excised leaves and petioles	Sprayed with a spore suspension three times at 24 h intervals	Leaf spot, chlorosis; petiole blight (darkening and necrosis); petioles were affected more than leaves	[43]
USA and England	Roots	Roots	Gnotobiotic culture: the spore suspension was pipetted into a vertical hole in the agar near the plant roots, which were growing rooted in tubes with agar. Slant-board culture: Pieces of alfalfa stem colonized with *P. medicaginis* were placed in contact with roots, which were spread fanlike in the slant culture board; Greenhouse culture: wounded and nonwounded roots were respectively immersed for 1 h in a spore suspension of 2 × 10^6^ spores/mL	Gnotobiotic culture: blackish, dry necrosis and tissue collapse were observed at or around the infection site. Withering, wilting, chlorosis, and reddening of foliage occurred on plants with severely diseased roots. Slant-board culture: Roots were often collapsed, with necrosis extending above and below the inoculation site. Greenhouse culture: necrosis occurred, especially on lateral roots.	[47]
USA	Crown	Crown of seedling and old plant, and stubble	Slant-board culture: placing an infested cloth square or applying spores with *P. medicaginis* to the surfaces of wounded crowns Greenhouse culture: crown was inoculated by using the infested needle technique; stubble inoculation with a toothpick infested with a mass of conidia	Crowns: intact crowns produced no rot symptoms; wounded crowns produced stub dieback and necrosis around wound site in slant-board and greenhouse culture. Stubble: black necrosis was observed initially; necrosis advanced down the stub, and stopped at the next node	[46]
Canada	Leaves	Leaves	5 mm mycelial disks were placed on detached leaves	Mean lesion size on detached leaves of 18 alfalfa cultivars 8 d after inoculation ranged from 2.5–8.2 mm in diameter	[26]
Germany	Asymptomatic leaves, petioles and stems	Leaves	Spraying spore suspension	Larger lesions and confined necrotic spots appeared on leaves within 2 weeks of spray-inoculation	[33]
England and USA	Leaves, stems, crowns roots and seeds	Roots and crowns	Roots (growth chamber): roots were stabbed with a needle, and a piece of colonized stem with *P. medicaginis* was placed on non-wounded and wounded roots; Roots (greenhouse): Roots were wounded by abrasion with a file, then immediately immersed for 1 h in a spore suspension of 2 × 10^6^ spores/mL; Crowns: crowns were inoculated by stabbing with a needle, then contaminated with spores	Roots (growth chamber): lesions in the wounded inoculated roots were longer than those in non-wounded inoculated roots; Roots (greenhouse): necrosis lesion Crowns: internal necrosis of the stem and upper taproot; extensive discoloration of the vascular tissues	[45]
America	Leaves	Leaves	Spraying spore suspension at 1.6 × 10^6^ spores/mL	Chlorosis and spots	[40]
	Leaves and stems	Leaves and stems	Spraying spore suspension	Lesions were apparent at 4 days postinoculation (dpi); At 12 dpi, many dark brown lesions with chlorotic halos were noted on leaves, occasionally killing entire trifoliate leaves and progressing approximately 1 cm down the stem	[52]
Tunisia	Leaves and stems of *M. truncatula*	Leaves and roots	Leaves: spraying spore suspension at 1.0 × 10^6^ spores/mL; Roots: conidia suspensions were depositing on roots (3 mm from the root apex) by using an in vitro inoculation method.	Leaves: necrosis and yellowing; Pycnidium production was observed on dead and dying foliar tissues; Roots: collar rot and brown discoloration; Pycnidium production was observed on collars and roots	[41]

## Data Availability

Not applicable.

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
