# Peer review of "Alfalfa Spring Black Stem and Leaf Spot Disease Caused by Phoma medicaginis: Epidemic Occurrence and Impacts"

_microorganisms, 2024, doi:10.3390/microorganisms12071279_

Round 1
Reviewer 1 Report
Comments and Suggestions for Authors
Final report microorganisms -3004474
Alfalfa spring black stem and leaf spot disease caused by Phoma medicaginis: epidemic occurrence and impacts
The primary focus of this study is on the taxonomic categorization of the pathogen, its host range, and the symptoms and harm associated with ASBS. Additionally, this review offers basic information regarding the propagation of infection, conditions that are predisposed to epidemics, resistance mechanisms, control methods, and ways to continue research.
We can NOT publish a review article with only one table. Please add more tables please
We can NOT publish a review article with only 2 figures table. Please more figures please.
Under section 2.1 symptoms. Please add a figure to show us the fungus morphology
Under section 2.2, Culture characteristics. Please add a figure to show us the Cultural characteristics
Under section 4, symptoms. Please add a figure to show us the symptoms
Section 8.1 Host resistance must be expanded.
Section 8.3 Biological control must be expanded.
Section 8.4 Cultural Practice must be expanded.
Section 8.5 Chemical control must be expanded.
To make the paper better, I have some comments that I need to make, which are as follows:
Do not use acronyms in the captions of the figures or in the footnotes; instead, ensure that each figure can be comprehended on its own.
Every figure must be self-explanatory.
It is imperative that you make certain that all tables are self-explanatory and that any abbreviations are explained in their entirety.
I would suggest incorporating additional references from the years 2023 and 2024, despite the fact that all of the currently provided references are sufficient.
I noticed this when I was looking through the journals. One was written in its entirety, while others were written in a more condensed style.
You are required to adhere to the format, which consists of either the full name of the journal or a condensed version of it. This is a requirement of the journal style. It is imperative that you take the time to read the instructions that have been provided to the authors.
In the process of supplying references for textbooks, it is recommended that the page numbers of the textbook themselves be noted.
It is possible that this paper could be approved with some minor modifications if the authors were the ones who made all of the suggestions that were mentioned above.
To make sure that all of my suggestions have been taken into consideration, I will need to make one more revision.
Comments on the Quality of English Language
Moderate editing of English language required
Author Response
|
Reviewer 1: |
||||
|
Comment ID |
Reviewer comment |
Author response |
||
|
R1.1 |
The primary focus of this study is on the taxonomic categorization of the pathogen, its host range, and the symptoms and harm associated with ASBS. Additionally, this review offers basic information regarding the propagation of infection, conditions that are predisposed to epidemics, resistance mechanisms, control methods, and ways to continue research. |
Thank you for the synopsis. No action needed by the authors. |
||
|
R1.2 |
We can NOT publish a review article with only one table. Please add more tables. |
Thanks. No Tables are added, but it is not from unwillingness to meet the reviewer’s request. The reviewer has not indicated specifically what is missing, Reviewer 2 has not raised any issue. It was not evident to us on reflection where in the MS a Table could be beneficial, and we know of no publishing convention that says a review must contain Tables. Rather the insertion of Tables or not is decided on a case by case basis to enhance conciseness of information collation.
|
||
|
R1.3 |
We can NOT publish a review article with only 2 figures table. Please more figures . Under section 2.1 symptoms. Please add a figure to show us the fungus morphology Under section 2.2, Culture characteristics. Please add a figure to show us the Cultural characteristics Under section 4, symptoms. Please add a figure to show us the symptoms.
|
Four figures have been added as follows: 1. Figure 1. The morphological characteristics of Phoma medicaginis varieties medicaginis (A and C, at left) and macrospora (B and D, at right) cultured at 25°C (A and B, above) and 5°C (C and D, below). Figure 2. Authors’ unpublished data from an experiment comparing growth of Phoma medicaginis at 25°C on different culture media. Methodology details for the experiment are given in Supplementary Information on line, S1. The photos for each culture medium are representative of 4 physical replicates and all depict the same isolate: A, cultured on oatmeal agar; B, potato carrot agar; C), potato dextrose agar; D, potato sucrose agar; E, malt extract agar; and D) czapek dox agar. Figure 3. Phoma medicaginis colony diameter after culture for 6 days on potato dextrose agar at A) 5°C; B) 10°C; C) 15°C; D 20°C: E)25°C; F) 30°C. Photos are from an unpublished experiment by the authors which included 4 replicates and 24 P. medicaginis strains. Figure 4. Symptoms associated with Phoma medicaginis infection of alfalfa leaves, stems and roots observed in the fields by the authors. A, small black spots on the leaf; B, “V” shaped necrotic areas on the leaf tip; C, necrotic spots on the leaf with concentric rings; D and E, stem blackening; F, root necrotic lesion and rot. |
||
|
R1.4 |
Section 8.1 Host resistance must be expanded. |
Thanks. We searched, but found only one additional reference on this topic: in addition to the enhanced expression of genes for chitinase activity, phenylalanine metabolism and photosynthesis activity, increased resistance to P. medicaginis infection also induced by inoculating the roots with AMF (Li et al., 2021). (Please see section 8.1) |
||
|
R1.5 |
Section 8.3 Biological control must be expanded. |
Thanks. The sentences below have been added to the revised manuscript in section 8.3: In addition, Sinorhizobium meliloti strain 10.16/R6 has been reported to be antagonistic to P. medicaginis and to decrease vegetative rotting length and plant mortality of M. truncatula plants infected with P. medicaginis by 65% and 80% [91]. Many species of the genus Bacillus have been used to control alfalfa disease, and they show broad-spectrum antifungal activity. The protective ability of B. licheniformis was attributed to its high chitinolytic activity and antifungal properties [88]. B. subtilis strain L194 also alleviates disease symptoms by reducing germination of P. medicaginis conidia [92]. B. subtilis can also control alfalfa root rot disease. For example, the B. subtilis strain CG-6 exerted an 87.33% growth inhibition against Fusarium oxysporum and the protective ability of B. subtilis was attributed to secreted antibacterial enzymes, siderophores and indoleacetic acid, and phosphate solubilization [93]. B. amyloliquefaciens against alfalfa anthracnose disease (Colletotrichum truncatum) can produce bacillomycin D and fengycin [94]. Nonribosomal peptide (NRP) surfactins, the largest class of Bacillus spp. antibiotics, were regarded as playing an important role in the antagonism of Bacillus spp. Towards P. medicaginis [89]. |
||
|
R1.6 |
Section 8.4 Cultural Practice must be expanded. |
Thanks. We searched, and located two additional references, now cited as follows: More cultural control strategies, including grazing, fertilizer application, rotations and seed health are also a strategy to manage and reduce alfalfa disease [13]. For example, green mature and crop rotation also significantly reduced alfalfa root rot and yield [98]. |
||
|
R1.7 |
Section 8.5 Chemical control must be expanded |
Thanks. Information below was added to the revised manuscript: Solamargine at 60 µM concentration and pH 7, caused a 50% growth inhibition of P. medicaginis [104]. |
||
|
R1.8 |
To make the paper better, I have some comments that I need to make, which are as follows: . |
Thanks. All Figure captions and footnotes have been edited to explain acronyms in full on first usage. |
||
|
R1.9 |
Every figure must be self- Do not use acronyms in the captions of the figures or in the footnotes; instead, ensure that each figure can be comprehended on its own explanatory. It is imperative that you make certain that all tables are self-explanatory and that any abbreviations are explained in their entirety. |
Thanks. Acronyms in figures are now explained in full. |
||
|
R1.10 |
I would suggest incorporating additional references from the years 2023 and 2024, despite the fact that all of the currently provided references are sufficient. |
Thanks. Relevant references are scarce but we did manage to find three: 1. Omidvari, M.; Flematti, G.R.; You, M.P.; Abbaszadeh-Dahaji, P.; Barbetti, M. J. Plant growth stage and Phoma medicaginis inoculum con centration together determine severity of Phoma black stem and leaf spot and consequent phytoestrogen production in annual Medicago spp. Plant Pathol. 2023, 72, 1463–1475. 2. Chen, J. X., Cai, R., Tang, L., Wang, D., Lv, R., and Guo, C. H. 2024. Antagonistic activity and mechanism of Bacillus subtilis CG-6 suppression of root rot and growth promotion in Alfalfa. Microbial Pathogenesis. 190: 106616. 3. Botkin, J.R.; Farmer, A.D.; Young, N.D.; Curtin, S.J. Genome assembly of Medicago truncatula accession SA27063 provides insight into spring black stem and leaf spot disease resistance. BMC Genomics. 2024, 25, 204. |
||
|
R1.11 |
I noticed this when I was looking through the journals. One was written in its entirety, while others were written in a more condensed style. You are required to adhere to the format, which consists of either the full name of the journal or a condensed version of it. This is a requirement of the journal style. It is imperative that you take the time to read the instructions that have been provided to the authors. |
Thanks. We have spent time looking at points for improvement of reference presentation and in doing so have paid attention to these two reviewer requests. |
||
|
R1.12 |
In the process of supplying references for textbooks, it is recommended that the page numbers of the textbook themselves be noted. |
|||
|
R1.13 |
It is possible that this paper could be approved with some minor modifications if the authors were the ones who made all of the suggestions that were mentioned above. To make sure that all of my suggestions have been taken into consideration, I will need to make one more revision. |
We have endeavoured to meet ALL reviewer requests. |
||
|
R1.14 |
Comments on the Quality of English Language. Moderate editing of English language required |
Prior to submission the English language was extensively revised by Professor Cory Matthew who is a native English speaker from New Zealand and currently Managing Editor of the Wiley journal Grassland Research. Professor Matthew is a little surprised to receive this comment but has respectfully taken a second look, even so, and made some more adjustments to the English grammar and syntaxes used. |
||
Reviewer 2 Report
Comments and Suggestions for Authors
Recommendations
L25 you could mention that biological control is promising (§ 8.3)
Introduction is focused on Alfalfa plant and not to the phytopathological disease (Phoma)
L52-53 Write a paragraph to mention the most important diseases of Alfalfa through the world and the infection level by Phoma among these diseases.
Described the symptoms that caused on Medicago plants by Phoma. Although you refer to the symptoms in paragraph 4. «Symptoms in infected alfalfa tissues», they should also be mentioned briefly in the introduction to highlight the phytopathological problem.
L105 “Culture media characteristics” or “Growth media characteristics”
References
For references see “Instructions for Authors” at “Manuscript Preparation”
…In the text, reference numbers should be placed in square brackets [ ], and placed before the punctuation; for example [1], [1–3] or [1,3]……
Comments on the Quality of English LanguageModerate editing of English language required. Text could be edited by a native speaker
Author Response
|
Reviewer 2: |
||
|
Comment ID |
Reviewer comment |
Author response |
|
R2.1 |
L25 you could mention that biological control is promising (§ 8.3) |
Thanks, “is promising” has been added at line 25. |
|
R2.2 |
Introduction is focused on Alfalfa plant and not to the phytopathological disease (Phoma) |
Thanks. We wanted to cover in the first paragraph of the introduction, the uses, importance, geographic distribution countries and planting areas of alfalfa globally. We feel this is relevant context. The second paragraph of introduction section is focused on Phoma medicaginis to let the readers get to the point of this manuscript, and in response to Reviewer 2 we have added additional material (highlighted yellow. |
|
R2.3 |
L52-53 Write a paragraph to mention the most important diseases of Alfalfa through the world and the infection level by Phoma among these diseases. |
Thanks. the informations below was added to the revised manuscript: Alfalfa leaf spot diseases including Leptosphaerulina spp., anthracnose disease, and Pseudopeziza spp., among others, occur throughout the entire growing season. In particular, alfalfa spring black stem and leaf spot disease (ASBS) caused by Phoma medicaginis, is a cosmopolitan soil-borne [18,19] and seed-borne disease [19,20]. The disease symptoms typically include leaf spot, black stem, seedling blight, and crown and root rot. |
|
R2.4 |
Described the symptoms that caused on Medicago plants by Phoma. Although you refer to the symptoms in paragraph 4. «Symptoms in infected alfalfa tissues», they should also be mentioned briefly in the introduction to highlight the phytopathological problem. |
Thanks. the sentence below was added to the revised manuscript. ASBS symptoms typically include leaf spot, black stem, seedling blight, and crown and root rot. |
|
R2.5 |
L105 “Culture media characteristics” or “Growth media characteristics” |
Thanks. “Cultural characteristics” is more suitable. Therefore, “Cultural characteristics” replace “Cultural characteristics” in revised manuscript. |
|
R2.6 |
References For references see “Instructions for Authors” at “Manuscript Preparation” |
We apologize for this oversight. The references are now organized in MDPI stylewith sequential numbers |
|
R2.7 |
Comments on the Quality of English Language Moderate editing of English language required. Text could be edited by a native speaker |
Prior to submission the English language was extensively revised by Professor Cory Matthew who is a native English speaker from New Zealand and currently Managing Editor of Grassland Research. Professor Matthew is a little surprised to receive this comment but has respectfully taken a second look, even so, and made some more adjustments to the English syntaxes used. |
Round 2
Reviewer 1 Report
Comments and Suggestions for Authors
Accept in present form. The authors did all my comments.
Author Response
|
Reviewer 1: |
||
|
Comment ID |
Reviewer comment |
Author response |
|
R1.1 |
The review paper microorganisms-3004474_R1 has been improved enough to be published in the journal |
Thank you for the help and suggestions. |
|
R1.2 |
Add the name of authors followed by the word unpublished |
Thanks. “from unpublished data by Yanru Lan” were add to the in the right place |
|
R1.3 |
Please check that all references are relevant to the contents of the manuscript. |
Thanks. There is a one-to-one correspondence between the literature and the contents in the text. The serial number of cited literature “[20]” was changed to “[21]” in line 54. |
|
R1.4 |
Spell the acronym of AMF |
arbuscular mycorrhizal fungi |
|
R1.5 |
Any revisions to the manuscript should be highlighted, such that any changes can be easily reviewed by editors and reviewers |
The revised contents were highlighted in yellow. |
|
R1.6 |
Based on Systematic Botany and Mycology Laboratory records (https://nt.ars- 73 grin.gov/fungaldatabases/index.cfm, accessed on 15 January 2020) [24]
Old date please see now if something changed
|
We visited again address, the content could be not found. So we deleted this sentence and information. |

Reviewer 2 Report
Comments and Suggestions for Authors
Dear Authors
The Review paper microorganisms-3004474_R1 has been improved enough to be published in the journal
Comments on the Quality of English Language
The English language of the text needs minor corrections
Author Response
|
Reviewer 2: |
||
|
Comment ID |
Reviewer comment |
Author response |
|
R1.1 |
Accept in present form. The authors did all my comments. |
Thank you for the help and suggestions. |
|
R1.2 |
Add the name of authors followed by the word unpublished |
Thanks. “from unpublished data by Yanru Lan” were add to the in the right place |
|
R1.3 |
Please check that all references are relevant to the contents of the manuscript. |
Thanks. There is a one-to-one correspondence between the literature and the contents in the text. The serial number of cited literature “[20]” was changed to “[21]” in line 54. |
|
R1.4 |
Spell the acronym of AMF |
arbuscular mycorrhizal fungi |
|
R1.5 |
Any revisions to the manuscript should be highlighted, such that any changes can be easily reviewed by editors and reviewers |
The revised contents were highlighted in yellow. |
|
R1.6 |
Based on Systematic Botany and Mycology Laboratory records (https://nt.ars- 73 grin.gov/fungaldatabases/index.cfm, accessed on 15 January 2020) [24]
Old date please see now if something changed
|
We visited again address, the content could be not found. So we deleted this sentence and information. |
